# Prediction Accuracy of Serial Lung Ultrasound in COVID-19 Hospitalized Patients (Pred-Echovid Study)

**DOI:** 10.3390/jcm10214818

**Published:** 2021-10-20

**Authors:** Juan Torres-Macho, Marcos Sánchez-Fernández, Irene Arnanz-González, Yale Tung-Chen, Ana Isabel Franco-Moreno, Mercedes Duffort-Falcó, Luis Beltrán-Romero, Santiago Rodríguez-Suaréz, Máximo Bernabeu-Wittel, Elena Urbano, Manuel Méndez-Bailon, Fernando Roque-Rojas, Elena García-Guijarro, Gonzalo García-Casasola

**Affiliations:** 1Internal Medicine Department, Infanta Leonor-Virgen de la Torre University Hospital, 28031 Madrid, Spain; anaisabel.franco@salud.madrid.org (A.I.F.-M.); mercedes.duffort@salud.madrid.org (M.D.-F.); 2Department of Medicine, School of Medicine, Complutense University, 28040 Madrid, Spain; arnanz.irene@gmail.com (I.A.-G.); manuelmenba@hotmail.com (M.M.-B.); ggcasasolaster@gmail.com (G.G.-C.); 3Internal Medicine Department, Doce de Octubre University Hospital, 28041 Madrid, Spain; marcossanferab@gmail.com; 4Emergency Department, Infanta Leonor-Virgen de la Torre University Hospital, 28031 Madrid, Spain; 5Emergency Department, La Paz University Hospital, 28046 Madrid, Spain; yale.tung.chen@gmail.com; 6Enfermera Isabel Zendal Emergency Hospital, 28055 Madrid, Spain; 7Internal Medicine Department, Virgen del Rocío University Hospital, 41013 Sevilla, Spain; luism.beltranromero@gmail.com (L.B.-R.); santiagorodriguezes@gmail.com (S.R.-S.); wittel@cica.es (M.B.-W.); 8Internal Medicine Department, Hospital Clínico San Carlos, 28040 Madrid, Spain; eurbano@gmail.com; 9Internal Medicine Department, Hospital Universitario Infanta Cristina, 28981 Parla, Madrid, Spain; fernando.roque@salud.madrid.org (F.R.-R.); egguijarro@salud.madrid.org (E.G.-G.)

**Keywords:** lung ultrasound, COVID-19, pneumonia, point-of-care ultrasound, prognosis

## Abstract

The value of serial lung ultrasound (LUS) in patients with COVID-19 is not well defined. In this multicenter prospective observational study, we aimed to assess the prognostic accuracy of serial LUS in patients admitted to hospital due to COVID-19. The serial LUS protocol included two examinations (0–48 h and 72–96 h after admission) using a 10-zones sequence, and a 0 to 5 severity score. Primary combined endpoint was death or the need for invasive mechanical ventilation. Calibration (Hosmer–Lemeshow test and calibration curves), and discrimination power (area under the ROC curve) of both ultrasound exams (SCORE1 and 2), and their difference (DIFFERENTIAL-SCORE) were performed. A total of 469 patients (54.2% women, median age 60 years) were included. The primary endpoint occurred in 51 patients (10.9%). Probability risk tertiles of SCORE1 and SCORE2 (0–11 points, 12–24 points, and ≥25 points) obtained a high calibration. SCORE-2 showed a higher discrimination power than SCORE-1 (AUC 0.72 (0.58–0.85) vs. 0.61 (0.52–0.7)). The DIFFERENTIAL-SCORE showed a higher discrimination power than SCORE-1 and SCORE-2 (AUC 0.78 (0.66–0.9)). An algorithm for clinical decision-making is proposed. Serial lung ultrasound performing two examinations during the first days of hospitalization is an accurate strategy for predicting clinical deterioration of patients with COVID-19.

## 1. Introduction

Despite substantial efforts to prevent the spread of coronavirus 19 disease (COVID-19), at the end of February 2021, over 115 million people worldwide had tested positive for SARS-CoV-2, and more than 2.5 million had died [1].

The wide variation in the symptoms of COVID-19 makes it difficult to predict the clinical course, and this complicates triage. Clinical experience has demonstrated significant heterogeneity in the course of SARS-CoV-2 infection: while some patients are asymptomatic or progress with mild symptoms, others develop severe acute respiratory distress syndrome with multiorgan failure and death [2].

Early detection of cases that are at high risk of progression to severe COVID-19 with an imminent risk of death is urgent and challenging [3].

Lung ultrasound (LUS) has been proposed as a suitable alternative to conventional radiological methods to diagnose COVID-19 pneumonia. Chest X-ray has shown a low sensitivity for detecting COVID-19 with false negatives in more than 40% of cases [4]. Computed tomography (CT) is a sensitive technique to diagnose and monitor the progression of COVID-19 pneumonia but its use is limited due to some significant drawbacks: availability, exposure to ionizing radiation, time-consumption or need to transfer the patient to the CT room. Conversely, LUS is portable, quick, repeatable, easy to learn, does not use ionizing radiation, and minimizes the need for moving the patient reducing the risk of cross-contamination and contributing to the safety of healthcare providers and patients [5].

LUS has a high sensitivity for diagnosing COVID-19 pneumonia and a good correlation with high-resolution chest computed tomography (HRCT) [6,7]. Moreover, the severity of LUS abnormalities is associated with clinical outcomes in different clinical settings [8,9,10]. However, most prognostic studies are based on one time-point LUS and serial ultrasound examinations could add prognostic information. In patients with an initial mild LUS pattern, LUS monitoring could anticipate clinical deterioration, which is not so uncommon in these patients. To date, there is a scarcity of data about serial LUS prognostic relevance or cut-off values that can help in clinical decision-making, nor solid indications on timing and approaches for monitoring patients.

The aim of this study is to analyze the prognostic accuracy of serial lung ultrasound examinations in patients admitted to hospital due to COVID-19.

## 2. Materials and Methods

This was a multicenter prospective observational study performed at six University Hospitals in Spain that recruited patients older than 18 years admitted to hospital due to SARS-CoV-2 infection confirmed using reverse-transcription polymerase chain reaction (PCR) of SARS-CoV-2 RNA from nasopharyngeal swabs. Exclusion criteria were patients < 18 years, patients with invasive mechanical ventilation at inclusion, patients with previously known interstitial lung disease, and patients who refused to participate. Patients were recruited using convenience sampling based on sonographer availability.

Patient demographics and clinical data were collected at hospital admission. Two ultrasound examinations were planned to be performed, the first one within 48 h after hospital admission (SCORE-1) and the second on day 3 or 4 (72–96 h) after the first examination (SCORE-2).

The extent and severity of the ultrasound findings were determined using a 10-zones protocol (Figure 1). Each intercostal space of the upper and lower parts of the anterior, lateral, and posterior regions of the left and right chest wall were examined. For each of the 10 zones, a score from 0 to 5 was given depending on sonographic findings: A pattern (0 points), isolated B-lines defined as less than 3 in a 3 s clip (1 point), more than 3 B lines during a 3 s clip (2 points), coalescent B-lines (3 points), small (<1 cm) subpleural consolidations (4 points) and consolidations > 1 cm (5 points). The total LUS score was calculated by summing the scores of all 10 zones (range of possible scores: 0–50). The presence of pleural effusion was registered. Appendix A show LUS score images and clips.

Ultrasound scans and scoring were performed by an operator with experience in LUS (at least 50 previous supervised examinations) who was not in charge of the patient. A 3–5 Mhz convex probe or a 5–13 Mhz linear probe, was used according to the patient’s physical characteristics and the region to explore. Handheld devices (V-scan or Butterfly) were allowed. Appendix A shows the number of patients recruited in each center and the ultrasound machine used to perform LUS examinations.

Together with the ultrasound evaluation, blood samples were obtained according to local clinical protocols determining blood leukocyte and lymphocyte count, D-dimer, ferritin, and C reactive protein (CRP). The closest determination to ultrasound examinations was registered.

In this study we analyzed the association between adverse outcomes and the severity of pulmonary involvement evaluated using LUS (as assessed by the total score) in the first 48 h after admission (SCORE-1), at 72–96 h after the first examination (SCORE-2) and the difference among both scores (DIFFERENTIAL SCORE). The primary outcome was the composite of death or the need for invasive mechanical ventilation during hospitalization.

### 2.1. Ethics

The study was conducted in accordance with the Declaration of Helsinki. Due to infective risk for healthcare workers, oral informed consent was obtained from each enrolled patient and this information (date of information and patient’s oral consent) was registered in the clinical history. An informed consent sheet was given to the patient. Study procedures did not delay or modify any diagnostic or therapeutic intervention. The study protocol was approved by the Research Ethics Committee of Hospital Universitario Puerta de Hierro (PI: 116/20).

### 2.2. Statistical Analyses

Data were managed using REDCap electronic data capture tools hosted at the Ideas for Health Association. REDCap (Research Electronic Data Capture) is a secure, web-based software platform designed to support data capture for research studies [11].

Continuous variables were expressed as means and standard deviations. To compare the means of two groups of independent samples, Student’s *t*-test was applied. To establish the best cut-off values of risk intervals, we determined the calibration of SCORE-1 and SCORE-2, by comparing their predicted probability of developing the endpoint to the observed endpoint in the whole cohort. Then we calculated the Hosmer–Lemeshow goodness-of-fit test (H–L) of the risk strata and constructed the calibration curves. For this purpose, we divided the cohort into probability risk deciles, quartiles, and tertiles in order to assess the behavior of the different risk strata; finally, we opted to divide the cohort into risk tertiles, as this division yielded the best fitness.

In order to evaluate the discrimination power of both scores, we applied the point scoring system, determining risk scores for each participant, and calculating the area under the receiver operating characteristic curve (AUC-ROC).

Finally, we determined the sensitivity (Ss), specificity (Sp), positive and negative predictive values (PPV and NPV, respectively) of the three risk groups differentiated by SCORE-1 and SCORE-2. For this purpose, we assumed the development of the main event (invasive mechanical ventilation or death) as the absolute truth criterion.

We also performed the same analysis with the numerical difference obtained between SCORE-2 and SCORE-1 (DIFFERENTIAL SCORE). For this purpose, we recoded the values obtained by adding to all values the difference between the lowest value obtained (which was −25) and converting it to zero (assuming that the patient with the maximum ultrasound improvement obtained in the second ultrasound compared to the first one, could be considered as the one with the lowest risk in the cohort).

Statistics were performed using the IBM SPSS Statistics for Windows, version 22.0 (IBM Corp., Armonk, NY, USA), and Epidat 3.1 computer packs. A *p* < 0.05 was considered significant. The minimum sample size was estimated as 360 patients (accepting an alpha risk of 0.05 and a beta risk of 0.2 and expecting an incidence of the primary endpoint of 15%).

## 3. Results

Between August 2020 and March 2021, 469 patients were included. Two hundred and fifteen patients (45.8%) were male. The median age was 60 yrs (IQR 45–72). The patient’s flowchart is shown in Figure 2. Baseline demographic and clinical characteristics are shown in Table 1. At admission, the patient’s respiratory situation was as follows: 116 patients (24.7%) had an adequate oxygen saturation breathing ambient air, 313 patients (66.7%) were receiving oxygen using nasal cannulas and 37 patients (7.8%) were on noninvasive mechanical ventilation or high-flow oxygen. The median time from the onset of symptoms to hospital admission was 7 days (IQR 4–9) and the median length of hospital stay was 7 days (IQR 5–10).

### 3.1. Lung Ultrasound Findings

All patients were evaluated with LUS during the first 48 h after admission to hospital and 284 (60.5%) patients received a second LUS 72 h after the first one. Median LUS scores in the first and second examination were 18 (IQR 12–24) and 17 (IQR 11–24), respectively.

The number of lung areas affected and the degree of aeration detected in both examinations are shown in Figure 3. The analysis of regional scores shows that the inferior-posterior regions were more frequently involved. Apart from the A pattern (27.2% of all the evaluated zones), confluent B-lines were the most common finding (22.4% of all the evaluated zones), and they were seen in 345 patients (73.6%). Subpleural consolidations were observed in 11.6% of all the evaluated zones and were present in 230 patients (49%) and large consolidations were detected in 2.3% of all the evaluated zones and they were present in 148 patients (31.6%). Pleural effusion was observed in 15 patients (3.1%). LUS performed at 72 h after the first one showed an increase in LUS score in 129 patients (45.4%).

### 3.2. Outcomes

Fifty-one patients reached the primary endpoint (37 patients required admission to ICU for invasive mechanical ventilation and 22 patients died). Among patients who received serial ultrasound examinations, 15 of them (5.2%) reached the endpoint.

The mean predicted probability of developing the main endpoint was 9.1% in SCORE-1 (range 4.4% to 24%) and 5% in SCORE-2 (range 0.8% to 29%). When divided into probability risk tertiles, the lowest risk stratum included patients with 0–11 points, the intermediate—those with 12–24 points, and finally the highest risk group included those patients with 25 or more points in both scores.

Both scores obtained a high calibration in the H–L goodness of fit test as well as in the calibration curves (Table 2 and Appendix A). SCORE-2 showed a higher discrimination power than SCORE-1 (AUC-ROC 0.72 (0.58–0.85) vs. 0.61 (0.52–0.7), respectively).

In the same way, the mean predicted probability of developing the main endpoint was 4.9% in the DIFFERENTIAL SCORE (range 0.2% to 39%). When divided into probability risk quartiles, the two lowest risk strata included all patients in whom ultrasonography had improved (difference between SCORE-2 and SCORE-1 ranging from −25 to 0), the third quartile included those whose ultrasonography worsened moderately (difference ranging 1–4 points), and the fourth quartile the remaining patients (difference 5 or more points). Hence, we divided the DIFFERENTIAL SCORE into these three probability risk groups (patients with ultrasonography improvement, those with moderate worsening, and those with further sonography worsening). The DIFFERENTIAL SCORE obtained a high calibration in the H–L goodness of fit test as well as in the calibration curves showing a higher discrimination power than SCORE-1 and SCORE-2 (AUC-ROC 0.78 (0.66–0.9)) (Figure 4 and Appendix A).

Finally, the diagnostic values of SCORE-1, SCORE-2, and DIFFERENTIAL SCORE are detailed in Table 3. For this purpose, we have detailed S, Sp, PPV, and NPV of the lowest risk strata and the highest risk strata. The highest PPV was found in the group of patients with a SCORE-1 ≥ 25 points; whereas the highest NPV was observed both in those patients with a SCORE-2 < 12 points, and those with a DIFFERENTIAL SCORE ≤ 0. According to these results, a practical algorithm that is easy to apply in clinical practice is proposed (Figure 5).

## 4. Discussion

In this study, we analyzed the accuracy of serial LUS monitoring lung injury to define the severity and prognosis of COVID-19 patients admitted to hospital. We found that combining LUS score at admission and the absence of progression at 72 h allow us to rule out death or the need of mechanical ventilation in a significant proportion of patients and in the same way a high initial score or a significant increase in the second examination score detected patients with a high risk of death or the need of mechanical ventilation. To our knowledge, this is the first multi-centric and the largest study dealing with this topic.

Although pulmonary CT is the most reliable imaging test for lung injury detection in patients with COVID-19, its routine use for this purpose is not possible in most hospitals. Lung ultrasound has shown in different studies to be more sensitive than chest X-rays in detecting lung lesions in COVID-19 [12,13,14] and it has a high level of agreement with CT [15].

The lesions that can be seen in SARS-CoV-2 pneumonia on LUS examination are well defined [16]. As the disease progresses, the sonographic appearances of pleural line irregularities are more intense, and B lines artifacts and subpleural consolidations increase in frequency and size [17,18]. In our study, we have followed a simplified examination protocol derived from a previous one proposed by Soldati et al. [19]. In our experience, in a significant proportion of patients, it is difficult to obtain a good view of the axillary region and the left antero-inferior sector which is usually occupied by the heart. Unfortunately, there is no unanimity of criteria in this regard and other researchers used other exploring protocols [20,21].

Regarding ultrasound lesions characteristics and their distribution in our patients, they did not differ from previous reports [16,22]. In the vast majority of patients, there was a bilateral involvement, with an evident predominance of interstitial lesions (B lines) and the most affected regions were the posteroinferior ones. In contrast to previous studies, we found a higher incidence of subpleural consolidations [23]. Pleural effusion was detected only in a minority of patients, thus confirming previous reports [16].

We have found that a score > 25 points within 48 h after hospital admission and after three days of hospitalization, discriminates with a high accuracy which patients are at a higher risk of death or invasive mechanical ventilation. On the other hand, an LUS score < 12 points had a very high negative predictive value of unfavorable evolution. If no progression of lung lesions was observed among serial ultrasound examinations, the negative predictive value was 99%. These findings are of undoubted clinical importance since they allow us to perform a better stratification of patient’s severity adapting clinical management: (1) to provide closer monitoring of patients with higher scores or even treat intensively, and (2) to identify patients with a good prognosis and to plan early discharges which is essential to optimize health resources in a pandemic period.

According to our results, previous reports showed that a high LUS score at admission accurately predicts poor outcomes but most of them have been carried out in a single center with a relatively small number of patients included [8,9,10,24,25,26,27]. Our study was multicentric and we have included a substantially greater number of patients. Very recently, a multicenter study protocol with similar objectives to ours has been proposed, but it is still ongoing and the results have not been published [28].

Three prospective studies analyzed the prognostic value of LUS using serial examinations in COVID-19 patients in hospital wards. All these studies were single-center studies and with a significantly smaller sample size than ours.

Rubio-Garcia al. analyzed the prognostic LUS value performed on admission, between 48 and 72 h later, and the day prior to discharge. The primary endpoint was in-hospital death and/or admission to the intensive care unit. A total of 130 patients were included. An LUS score over 75 percentile, showed a significantly higher proportion of in-hospital death and/or admission to the intensive care unit and a longer length of stay. A baseline LUS score of 22 was the point of maximum sensitivity for the primary endpoint (sensitivity = 76.9%; specificity = 62.1%; AUC = 0.693). Contrary to our results, the score did not change over the first 72 h of hospital stay, suggesting that it is fully informative upon admission [29].

Perrone et al. investigated the association between the LUS score and a combination of high-flow oxygen support, intensive care unit admission, or 30-day mortality in hospitalized COVID-19 patients. Fifty-two patients underwent LUS on admission and before their discharge. A median LUS score higher than 24 on admission was associated with an almost 6-fold increase in the odds of worsening (OR, 5.67; 95% CI, 1.29–24.8; *p* = 0.02). Second LUS examination was performed only in 24 patients and the proportion of patients with more severe disease decreased significantly, but due to the study design, this information was not useful in prognosis evaluation [18].

Casella et al. showed a correlation of total LUS score after 72 h with intensive care admission or death suggesting that ultrasonographic monitoring accurately reflects disease progression. A cut-off value of 17 accurately predicted the primary outcome (sensitivity 89%; specificity 85%) [23]. These results are in line with ours, but we included the difference between the first and second score which may reflect lung injury progression more precisely than an isolated score.

### Limitations

We did not analyze correlations between LUS and CT due to the study design. Although lung ultrasound score has good reproducibility (k 1⁄4 0.95) (20) and all the investigators who performed LUS had extensive prior experience, this technique is operator-dependent and could have influenced the final results. Another limitation is that examiners were not blinded to clinical severity. The second LUS examination was performed after a short interval since the first examination (less than one week). As only patients with a mild to moderate COVID-19 were included, a significant proportion were discharged home without the possibility to perform a second LUS following the inclusion. Moreover, a significant proportion of second ultrasound examinations (11.4%) were not performed because investigators were not available on day 3 and 4 after admission (mainly on weekends) and the impact of this high proportion of lost-to-follow-up patients on the prognostic performance of LUS is difficult to estimate, limiting the external validation of the proposed algorithm. Given that the median days of symptoms before admission was seven, the proposed algorithm should be used with caution in patients with a low-risk LUS and few days of symptoms.

## 5. Conclusions

The serial LUS protocol and algorithm we propose could be a valuable tool in medical wards to predict the probability of clinical worsening in patients with SARS-CoV-2 pneumonia, helping to identify the most appropriate therapeutic and management pathway for patients, and to monitor the disease with a simple and innocuous technique that can be performed at the patient’s bedside. This may facilitate the selection of patients who need closer monitoring and a more intense therapeutic approach. Moreover, our protocol can help to identify low-risk patients with a good prognosis that can be discharged earlier.

## Figures and Tables

**Figure 1 jcm-10-04818-f001:**
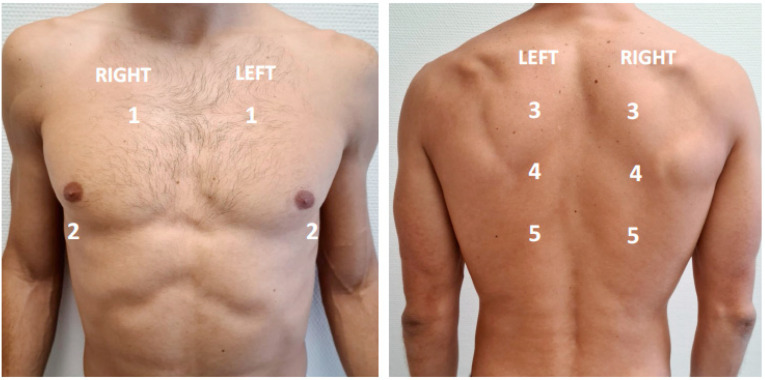
Lung ultrasound examination protocol using 10 zones.

**Figure 2 jcm-10-04818-f002:**
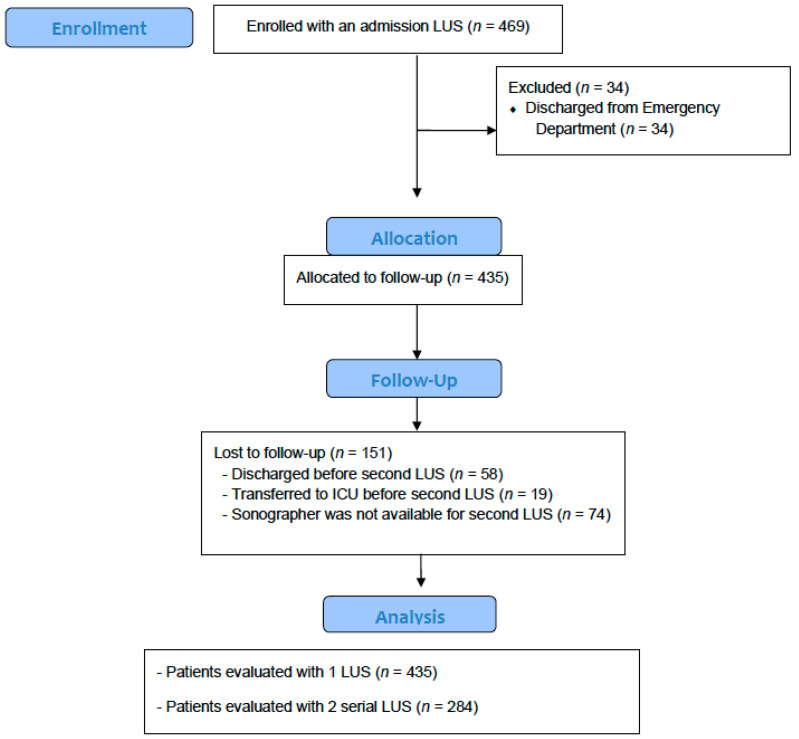
Consort 2010 flow chart of patients with COVID-19 admitted to Spanish hospitals and included in the PREDECHOVID study. LUS: lung ultrasound; ICU: intersive care unit.

**Figure 3 jcm-10-04818-f003:**
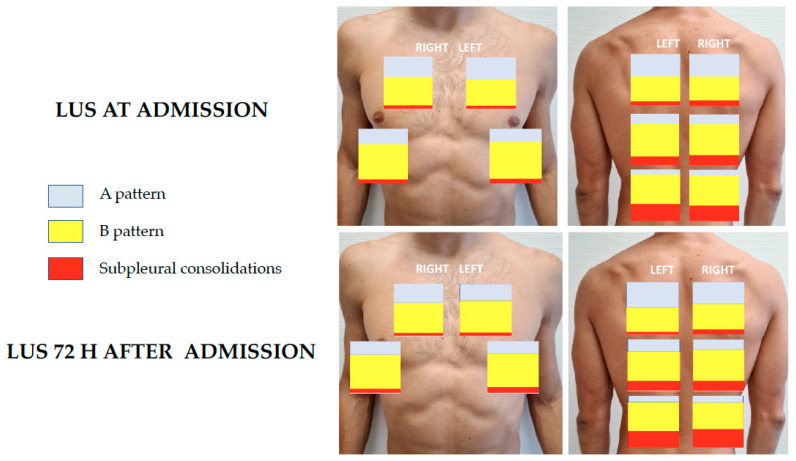
Distribution of LUS findings at admission and 72 h later.

**Figure 4 jcm-10-04818-f004:**
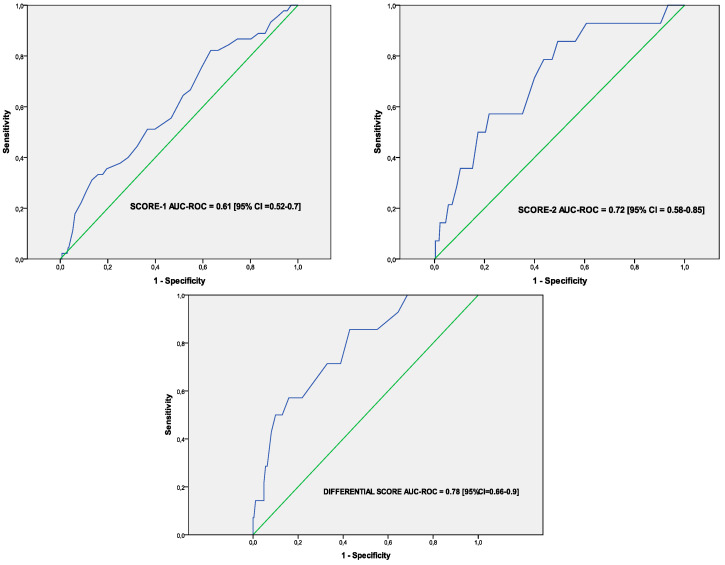
Discrimination power by means of the receiver operator characteristic and its area under the curve (AUC-ROC) of SCORE-1, SCORE-2 and DIFFERENTIAL SCORE.

**Figure 5 jcm-10-04818-f005:**
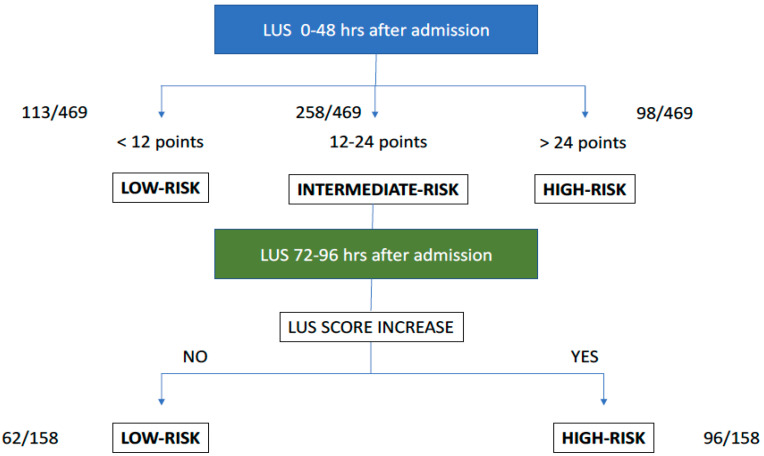
Suggested algorithm using serial LUS.

**Table 1 jcm-10-04818-t001:** Baseline demographic and clinical characteristics of patients.

Variables	*n* = 469
Age (yrs, SD)	60.5 (16.2)
Gender (male, %)	215 (45.8)
Race (n, %)	
Caucasic	299 (63.7)
Hispanic	131 (27.9)
Other	39 (8.3)
Comorbidities (*n*, %)	
Hypertension	199 (42.4)
Dyslipidemia	143 (30.4)
Diabetes	94 (20)
Obesity	135 (28.7)
Smoker	105 (22.3)
Cardiopathy	74 (15.7)
COPD	30 (6.3)
Asthma	37 (7.8)
Cerebrovascular disease	22 (4.6)
Analytical parameters	
Haemoglobin (g/dL, SD)	13.8 (1.6)
Total lymphocites (SD)	1.059 (1.038)
Platelets (10^9^/L, SD)	229.4 (102.2)
D-dimer (ng/mL, SD)	1277.3 (3263.2)
LDH (U/L, SD)	308.1 (120)
C-reactive protein (mg/L, SD)	62.3 (72.5)
IL-6 (pg/mL, SD)	39.8 (19.8)
Ferritin (ng/mL, SD)	714.1 (731)
Respiratory status at admission (n, %)	
Respiratory rate (Mean, SD)	19.5 (5.5)
No oxygen need	119 (25.3)
O2 nasal cannula	313 (66.7)
High flow/NIMV	37 (7.8)
Pa/FiO2 (Mean, SD)	440.5 (25,4)
Medical treatment (n, %)	
Remdesivir	45 (9.5)
Corticosteroids	384 (81.8)
Tozilizumab	27 (5.7)
Outcomes	
IMV	37 (7.8)
Death	22 (4.6)

COPD: Chronic obstructive pulmonary disease; LDH: Lactate dehydrogenase; IMV: invasive mechanical ventilation.

**Table 2 jcm-10-04818-t002:** Calibration of two point-of-care ultrasonographic scores by comparison of their risk strata classification predictions, and observed endpoint (orotracheal intubation or death) in a multicenter cohort of in-hospital patients with COVID-19 in Spain.

Index	Predicted Probability for Developing End-Point (95% CI) #	Observed End-Point †	Hosmer–Lemeshow Test: Chi-Square Test (Degrees of Freedom)
**SCORE-1** ¶			
0–11 points	6% (5.8–6.04%)	6 (5.3%)	0.022
12–24 points	9.3% (9.1–9.5%)	23 (9.1%)	*p* = 0.9
≥25 points	15% (14.5–15.5%)	16 (16.3%)	
**SCORE-2** *			
0–11 points	1.6% (1.5–1.7%)	1 (1.2%)	0.00
12–24 points	4% (3.8–4.2%)	5 (3.8%)	*p* = 0.993
≥25 points	11% (9.9–12%)	8 (11.9%)	
**DIFFERENTIAL SCORE** ⁂			
(−25 to 0 points)	2% (1.9–2.2%)	2 (1.3%)	0.63
(1–4 points)	4.4% (4.3–4.6%)	4 (6.6%)	*p* = 0.427
≥5 points	12% (9.8–14%)	8 (11.9%)	

† End-point: orotracheal intubation or death; CI: Confidence Interval; # Obtained by logistic regression modeling; ¶ Point of care ultrasound performed in the first 24 h of hospital stay; * Point of care ultrasound performed after 72 h and before 96 h of hospital stay. ⁂ DIFFERENTIAL SCORE: difference between SCORE-1 and SCORE-2.

**Table 3 jcm-10-04818-t003:** Comparison of sensitivity, specificity, positive predictive value, and negative predictive values, in the detecting of the endpoint (orotracheal intubation or death), between two point-of-care ultrasonographic scores. For each score and their combinations, the values obtained for the lowest-highest risk-strata are detailed.

Index	Sn	Sp	PPV	NPV
**SCORE-1 < 12 points**	87% (74–94%)	25.5% (21.5–30%)	11% (8–15%)	95% (89–97.5%)
**SCORE-1 ≥ 25 points**	35% (23–50%)	80% (76.4–84%)	16% (10–25%)	92% (89–94%)
**SCORE-2 < 12 points**	93% (68–99%)	31% (26–37%)	6.6% (4–11%)	99% (94–99.8%)
**SCORE-2 ≥ 25 points**	57% (32–79%)	78% (73–83%)	12% (6–22%)	97% (94–99%)
**SCORE-1 and SCORE-2 < 12 points**	98% (88–99.6%)	9.4% (7–12.6%)	10% (7.7–13.5%)	97.6% (87–99.6%)
**SCORE-1 and SCORE-2 ≥ 25 points**	4.4% (1.2–15%)	93% (90–95%)	6.5% (2–2.21%)	90% (87–93%)
**DIFFERENTIAL SCORE < 0 #**	86% (60–96%)	57% (51–63%)	9.4% (5.4–15.7%)	99% (95.5–99.6%)
**DIFFERENTIAL SCORE ≥ 5 points**	57% (33–97%)	78% (73–83%)	12% (6–22%)	97% (94–99%)

# DIFFERENTIAL SCORE: Difference between SCORE-2 and SCORE-1. Sn: Sensitivity; Sp: Specificity; PPV: Positive predictive value; NPV: Negative predictive value.

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
