# Peer review of "Prediction Accuracy of Serial Lung Ultrasound in COVID-19 Hospitalized Patients (Pred-Echovid Study)"

_jcm, 2021, doi:10.3390/jcm10214818_

Round 1
Reviewer 1 Report
1. This Spanish cohort is large and data have been collected across multiple sites, while most (6/7) are located in the same city (Madrid). Furthermore, the rationale for using lung ultrasound to predict worsening in COVID patients is solid, as correctly exposed by the authors in the introduction (lines 62-65). However, this idea is not new, and several reports with similar objectives and methods have been published already (see for instance Xing et al., Critical Care (2021) and Casella et al., Eur J Intern Med. (2021).
2. Thus, it is interesting to confront this report with what has been published previously by others. With regards to the main conclusions of the study:
-
In the proposed study, the predictive performance of LUS assessed at days 1-2 following hospital admission is low, with an AUC-ROC = 0.61 [0.52-0.7]. This number suggests against a significant clinical interest of monitoring LUS at baseline. Of note, the 95% IC has a lower margin of 0.52, indicating almost no added value of the test in comparison to chance alone.
-
The predictive performance of LUS assessed at later time points is somewhat higher (AUC-ROC 0.72 [0.58-0.85]), which is in line with the report by Casella et al. However, the clinical impact of performing the test seems rather limited and is not really stressed by the authors (see below).
-
Finally, one key idea that is put forward by the authors and that has not been extensively studied (to my knowledge) in the existing literature is that there could be some added predictive information obtained by comparing LUS values between days 1-2 and days 3-5. However this comparison is only possible in ~60% of the whole cohort, as 151 patients (over 435) were lost to follow-up between the first and the second examination. These patients may have gotten better (discharged home) or worse (transferred to ICU) before days 3-5, but in any case the impact of this high proportion of lost-to-follow-up patients on the prognostic performance of LUS is not easy to estimate.
Thus, in my opinion this report suggests that: 1) monitoring LUS at early time points is not useful to predict clinical worsening; 2) there is not enough data to conclude that serial monitoring has any added value.
3. There are some major methodological limitations that are not addressed, including:
- Whether examiners were blinded to clinical severity (probably not, as a difference between no oxygen vs. nasal cannulas vs. NIV or HFO is evident);
- How many patients were admitted to the ER with COVID yet did not have a LUS exam (and the reasons for this);
- What was the total patient follow-up (and was it planned in advance of data collection);
- What happened to the 34 patients that had an initial LUS exam yet were not followed-up because discharged from the ER.
4. With regards to the clinical use of LUS in this report, looking at Table 3 indicates that:
- All scores (SCORE-1, SCORE-2 and DIFFERENTIAL-SCORE) have a low PPV for death and/or intubation;
- All scores have a high NPV for death and/or intubation. This suggests that the clinical interest in performing the test would be to rule out clinical worsening if LUS is low at later time points. Looking at SCORE-2, the probability of a good outcome is 88% (1-PPV) if LUS is >25, 93.4% is LUS is >12, and 99% if LUS is <12. Interesting, but a limited gain of information (this is due to a low prevalence of the primary endpoint in the cohort and the statistical properties of the test).
This last point however seems like the one key finding of the paper: if patients assessed at days 3-5 have a low LUS, then worsening is unlikely. To interpret it correctly it would be mandatory in my opinion to provide the length of follow-up for all patients.
Following this, the suggested algorithm in Figure 5 does not appear to be backed by the data from Table 3. In the first line, the probabilities of meeting the primary endpoint are 11% in the « Low risk » and 16% in the « High risk » groups. Over the 258 patients in the « Intermediate risk » group, only 158 (61%) are evaluated at the next step, with probabilities of meeting the primary endpoint of 6.6% (« Low risk ») and 12% (« High risk »). Confusing...
5. Finally, 2 elements of the statistical analysis are not completely clear to me:
- How exactly were the score computed, or more specifically, how were the expected probabilities of the primary outcome calculated for each stratum of the scores;
- «The minimum sample size was estimated as 360 patients (...)» (lines 148-150): this number of patients was calculated (with an alpha risk of 0.05 and a Beta risk of 0.2 and expecting an incidence of the primary endpoint of 15%) to observe a difference of how much in which endpoint between which groups of patients?
In conclusion, I believe that this paper addresses an interesting question,
Author Response
- This Spanish cohort is large and data have been collected across multiple sites, while most (6/7) are located in the same city (Madrid). Furthermore, the rationale for using lung ultrasound to predict worsening in COVID patients is solid, as correctly exposed by the authors in the introduction (lines 62-65). However, this idea is not new, and several reports with similar objectives and methods have been published already (see for instance Xing et al., Critical Care (2021) and Casella et al., Eur J Intern Med. (2021).
It is not a new idea but previous studies were single-centre and the sample size was significantly lower.
- Thus, it is interesting to confront this report with what has been published previously by others.
In the discussion section, similar previous studies are mentioned and compared with ours.
- With regards to the main conclusions of the study:
- In the proposed study, the predictive performance of LUS assessed at days 1-2 following hospital admission is low, with an AUC-ROC = 0.61 [0.52-0.7]. This number suggests against a significant clinical interest of monitoring LUS at baseline. Of note, the 95% IC has a lower margin of 0.52, indicating almost no added value of the test in comparison to chance alone.
- The predictive performance of LUS assessed at later time points is somewhat higher (AUC-ROC 0.72 [0.58-0.85]), which is in line with the report by Casella et al. However, the clinical impact of performing the test seems rather limited and is not really stressed by the authors (see below).
- Finally, one key idea that is put forward by the authors and that has not been extensively studied (to my knowledge) in the existing literature is that there could be some added predictive information obtained by comparing LUS values between days 1-2 and days 3-5. However this comparison is only possible in ~60% of the whole cohort, as 151 patients (over 435) were lost to follow-up between the first and the second examination. These patients may have gotten better (discharged home) or worse (transferred to ICU) before days 3-5, but in any case the impact of this high proportion of lost-to-follow-up patients on the prognostic performance of LUS is not easy to estimate.
- This issue is mentioned in the limitations section: a significant proportion of second ultrasound examinations (11,4%) were not performed because investigators were not available on day 3 and 4 after admission (mainly on weekends) and the impact of this high proportion of lost-to-follow-up patients on the prognostic performance of LUS is difficult to estimate.
Thus, in my opinion this report suggests that: 1) monitoring LUS at early time points is not useful to predict clinical worsening; 2) there is not enough data to conclude that serial monitoring has any added value.
- There are some major methodological limitations that are not addressed, including:
- Whether examiners were blinded to clinical severity (probably not, as a difference between no oxygen vs. nasal cannulas vs. NIV or HFO is evident);
Examiners were not blinded to clinical severity. We have included this problem in the limitation section (page 11, line 344).
- How many patients were admitted to the ER with COVID yet did not have a LUS exam (and the reasons for this).
This data is not available. Patient´s inclusion depended on investigator´s availability.
- What was the total patient follow-up (and was it planned in advance of data collection);
Patients were followed up until hospital discharge.
- What happened to the 34 patients that had an initial LUS exam yet were not followed-up because discharged from the ER.
All patients included were admitted to hospital.
- With regards to the clinical use of LUS in this report, looking at Table 3 indicates that:
- All scores (SCORE-1, SCORE-2 and DIFFERENTIAL-SCORE) have a low PPV for death and/or intubation;
- All scores have a high NPV for death and/or intubation. This suggests that the clinical interest in performing the test would be to rule out clinical worsening if LUS is low at later time points. Looking at SCORE-2, the probability of a good outcome is 88% (1-PPV) if LUS is >25, 93.4% is LUS is >12, and 99% if LUS is <12. Interesting, but a limited gain of information (this is due to a low prevalence of the primary endpoint in the cohort and the statistical properties of the test).
This last point however seems like the one key finding of the paper: if patients assessed at days 3-5 have a low LUS, then worsening is unlikely. To interpret it correctly it would be mandatory in my opinion to provide the length of follow-up for all patients.
Length of follow up was until patient´s discharge. Median length of stay was 7 days and it was included in results section.
Following this, the suggested algorithm in Figure 5 does not appear to be backed by the data from Table 3. In the first line, the probabilities of meeting the primary endpoint are 11% in the « Low risk » and 16% in the « High risk » groups. Over the 258 patients in the « Intermediate risk » group, only 158 (61%) are evaluated at the next step, with probabilities of meeting the primary endpoint of 6.6% (« Low risk ») and 12% (« High risk »). Confusing...
The suggested algorithm was created using the probability of one patient to achieve the endpoint (5,3% if SCORE 1 was lower than 12 points and 16,3% if SCORE 1 was higher than 25 points).
It is true only 158 of 258 patients in the intermediate group received a second ultrasound examination. We included an explanation in the limitations section suggesting that an external validation is needed (page 12, lines 351-353).
- Finally, 2 elements of the statistical analysis are not completely clear to me:
- How exactly were the score computed, or more specifically, how were the expected probabilities of the primary outcome calculated for each stratum of the scores;
The probability of the development of the event was calculated individually by logistic regression, including the patient's index score as an independent variable.
These individual probabilities were then grouped into the pre-defined index score strata, obtaining the mean probability and corresponding measures of dispersion for developing the event in every risk stratum.
Finally, this mean probability for each stratum was compared with the actual number and percentages of patients who really developed the event in each index score stratum.
Reviewer 2 Report
I have read with interest this paper aimed at determining accuracy of LUS score to predict outcome in COVID-19.
Even if it repeats performance, I have found it of some value for daily practice.
However, the paper is affected by some major issue needing to be addressed.
- METHODOLOGY. The paper is somehow confusing and it is a little bit difficult to follow throughout the manuscript. While the 'introduction' section may benefit from a shortening, the 'methodology' section should be more systematically and extensively explained. All outcomes, endpoints, predictors and potential confounders should be clearer to the readers. Endpoints, and what the authors mean for SCORE-1 and SCORE-2 should be more extensively explained in a 'methods and measurements' dedicated section. To accomplish this, I strongly suggest to the authors to strictly follow the STROBE checklist, since it is an observational prospective cohort study.
- METHODOLOGY. There is no mention of sample size calculation nor power analysis. Is it the study enough powered or not? We should consider that only 51 patients out of more than 400 reached the primary composite endpoint. Please, provide a power analysis.
- Physiological parameters are missing. Oxygenation index and respiratory rate are fundamentals for prognosis COVID-19 respiratory failure, especially when integrated with LUS. Any comment about this?
- References should be updated. Please, eventually consider to quote the following paper underlining the role of LUS to predict non-invasive ventilation outcome: "Lung ultrasound predicts non-invasive ventilation outcome in COVID-19 acute respiratory failure: a pilot study" DOI: 10.23736/S0375-9393.21.15188-0.
Author Response
- METHODOLOGY. The paper is somehow confusing and it is a little bit difficult to follow throughout the manuscript. While the 'introduction' section may benefit from a shortening, the 'methodology' section should be more systematically and extensively explained. All outcomes, endpoints, predictors and potential confounders should be clearer to the readers.
Endpoints, and what the authors mean for SCORE-1 and SCORE-2 should be more extensively explained in a 'methods and measurements' dedicated section. To accomplish this, I strongly suggest to the authors to strictly follow the STROBE checklist, since it is an observational prospective cohort study.
We defined SCORE-1, SCORE-2 and DIFFERENTIAL SCORE in the text (page 3, line 94 and 95).
Strobe checklist was performed and included in suplemmentary material.
- METHODOLOGY. There is no mention of sample size calculation nor power analysis. Is it the study enough powered or not? We should consider that only 51 patients out of more than 400 reached the primary composite endpoint. Please, provide a power analysis.
Sample size calculation is mentioned in the statistics section: “The minimum sample size was estimated as 360 patients (accepting an alpha risk of 0.05 and a Beta risk of 0.2 and expecting an incidence of the primary endpoint of 15%)”. This estimation is in line with the one performed in a similar study (reference 28). Finally, we found an endpoint incidence of 12%, that´s why we decided to increase sample size from 360 to 436 to get to 50-55 events.
- Physiological parameters are missing. Oxygenation index and respiratory rate are fundamentals for prognosis COVID-19 respiratory failure, especially when integrated with LUS. Any comment about this?
The aim of this study was to analyze the predictive accuracy of serial LUS without any other parameter associated. We are planning to develop a prognostic score using clinical and laboratory parameters together with LUS.
- References should be updated. Please, eventually consider to quote the following paper underlining the role of LUS to predict non-invasive ventilation outcome: "Lung ultrasound predicts non-invasive ventilation outcome in COVID-19 acute respiratory failure: a pilot study" DOI: 10.23736/S0375-9393.21.15188-0.
This reference was included (number 27).
Round 2
Reviewer 1 Report
Some of my concerns/questions regarding the overall methodology are addressed by the authors in their reply, and some are also mentioned in the revised discussion, yet no substantial change in the analysis has been carried out.
I still find that this paper tackles an interesting question (« what is the predictive value of repeated lung ultrasounds to rule out/in ICU admission and/or death in COVID-19 patients? ») with some important methodological flaws (LUS assessment not blinded to disease severity; >35% of patients not assessed at secondary time point) and a rather limited demonstration of clinical utility (including low AUC at early time points and low probability of worsening in the highest risk group).
I still find that the most interesting point of the paper is that in patients with a low variation of the score between 2 assessments VPN for clinical worsening is very high, yet this finding is obscured by much less relevant results (in my opinion).
I agree that this is large cohort, and that the paper probably deserves to be made available to the community, and I will let the editor evaluate whether or not the Journal of Clinical Medicine is the most suited journal for this.
Author Response
Some of my concerns/questions regarding the overall methodology are addressed by the authors in their reply, and some are also mentioned in the revised discussion, yet no substantial change in the analysis has been carried out.
I still find that this paper tackles an interesting question (« what is the predictive value of repeated lung ultrasounds to rule out/in ICU admission and/or death in COVID-19 patients? ») with some important methodological flaws (LUS assessment not blinded to disease severity; >35% of patients not assessed at secondary time point) and a rather limited demonstration of clinical utility (including low AUC at early time points and low probability of worsening in the highest risk group).
All this points were answered in the first round. Methodological flaws (blinding, 35% of second LUS not performed) are mentioned in the text but they are not possible to change.
I still find that the most interesting point of the paper is that in patients with a low variation of the score between 2 assessments VPN for clinical worsening is very high, yet this finding is obscured by much less relevant results (in my opinion).
We believe that this point is highglighted in line 310 and 312 and included in the algorithm (72-96h hours LUS algorithm is based in this finding)